# Association of Serum Levels of Zinc, Copper, and Iron with Risk of Metabolic Syndrome

**DOI:** 10.3390/nu13020548

**Published:** 2021-02-07

**Authors:** Chia-Wen Lu, Yi-Chen Lee, Chia-Sheng Kuo, Chien-Hsieh Chiang, Hao-Hsiang Chang, Kuo-Chin Huang

**Affiliations:** 1Department of Family Medicine, National Taiwan University Hospital, 7 Zhongshan South Road, Taipei 100, Taiwan; biopsycosocial@gmail.com (C.-W.L.); jiansie@gmail.com (C.-H.C.); allanchanghs@gmail.com (H.-H.C.); 2Department of Family Medicine, College of Medicine, National Taiwan University, Taipei 100, Taiwan; omigodokuo@gmail.com; 3Department of Family Medicine, National Taiwan University Hospital Bei-Hu Branch, Taipei 108, Taiwan; MSAR1031@hotmail.com

**Keywords:** zinc, copper, iron, micronutrients, metabolic syndrome, insulin

## Abstract

The association between serum concentrations of zinc, copper, or iron and the risk of metabolic syndrome are inconclusive. Therefore, we conduct a case-control study to explore the relationship between serum levels of zinc, copper, or iron and metabolic syndrome as well as each metabolic factor and insulin resistance. We enrolled 1165 adults, aged ≥ 40 (65.8 ± 10) years in a hospital-based population to compare the serum levels of zinc, copper, and iron between subjects with and without metabolic syndrome by using multivariate logistic regression analyses. The least square means were computed by general linear models to compare serum concentrations of zinc, copper, and iron in relation to the number of metabolic factors. The mean serum concentrations of zinc, copper, and iron were 941.91 ± 333.63 μg/L, 1043.45 ± 306.36 μg/L, and 1246.83 ± 538.13 μg/L, respectively. The odds ratios (ORs) of metabolic syndrome for the highest versus the lowest quartile were 5.83 (95% CI: 3.35–10.12; *p* for trend < 0.001) for zinc, 2.02 (95% CI: 1.25–3.25; *p* for trend: 0.013) for copper, and 2.11 (95% CI: 1.24–3.62; *p* for trend: 0.021) for iron after adjusting for age, sex, personal habits, body mass index, and homeostatic model assessment insulin resistance. Additionally, the serum zinc, copper, and iron concentrations increased as the number of metabolic factors rose (*p* for trend < 0.001). This was the first study to clearly demonstrate that higher serum levels of zinc, copper, and iron were associated with the risk of metabolic syndrome and the number of metabolic factors independent of BMI and insulin resistance.

## 1. Introduction

Metabolic syndrome (MetS) is a cluster of abnormalities, including abdominal obesity, elevated blood pressure (BP), glucose intolerance, and dyslipidemia [1]. People with MetS are predisposed to diabetes mellitus (DM) and cardiovascular diseases. Due to the multifactorial interaction between genetic, metabolic, and environmental factors, diet and nutrition play important roles in the development of MetS [2]. Dietary micronutrients such as zinc (Zn), copper (Cu), and iron (Fe) are well known for their cooperation with numerous enzymes and their antioxidative functions. Consequently, these micronutrients are hypothesized to be involved in DM and MetS [3,4,5]. A few observational studies were designed to clarify the associations between serum concentrations of Zn, Cu, or Fe and DM, but the results were inconsistent [6,7,8]. In recent meta-analyses, it seems that serum levels of Cu and Fe tend to be positively associated with the risk of DM, while the association between serum levels of Zn and DM was inconclusive [9,10,11]. Supported by a large population-based, five-year cohort study, dietary intake of Cu and Fe was associated with a higher risk of new-onset DM [12].

Taking MetS and micronutrients into consideration, dietary intake of Zn was associated with a reduced risk of MetS [13], but dietary intake of Cu and Fe was associated with an increased risk of MetS [14]. However, little is known about the association between serum levels of Zn, Cu, or Fe and MetS while most of the results are not statistically significant [15,16,17]. Therefore, we conducted this large sample-size, case-control study to examine the association between Zn, Cu, and Fe levels and MetS as well as each metabolic factor.

## 2. Materials and Methods

### 2.1. Study Subjects

A total of 1165 apparently healthy subjects over 40 years old were enrolled in our study from 2007 to 2017 at a medical center in Northern Taiwan. They were invited for the health and nutrition survey in the outpatient clinic if they could comply with our study protocol. Information about age, sex, personal habits regarding cigarette smoking, alcohol consumption and exercise, current medications, and previous diseases was obtained by questionnaires through individual interviews after informed consent was signed. Personal habits of smoking and drinking were defined as binary categories. Current smokers were defined as those smoking for more than six months prior to participating in this study and labeled as “1”. Noncurrent smokers were defined as those who had quit for more than 12 months or had never smoked and labeled as “0”. Current alcohol drinkers were defined as those drinking more than one ounce of alcohol per week for six months and labeled as “1”. Noncurrent drinkers were defined as those who had quit for more than 12 months or had never drunk and labeled as “0”. Exercise habit was defined as a yes or no question by asking participants “do you have a regular exercise habit?” Weight and height were measured by a standard electronic scale and stadiometer. BP was measured by a sphygmomanometer. Waist circumference (WC) was measured by the same trained operator. Type 2 DM, hypertension, and hyperlipidemia, or other chronic diseases were defined based on a self-reported history or current medication use for those conditions. This study was approved by the Institutional Review Board of National Taiwan University Hospital (201511039RINA).

### 2.2. Definition of Metabolic Syndrome

Participants were considered to have metabolic syndrome if they met at least three of the following criteria: (1) WC equal to or greater than 90 cm in men or 80 cm in women; (2) systolic BP equal to or greater than 130 and/or diastolic BP equal to or greater than 85 mmHg or medication use for hypertension; (3) serum triglycerides equal to or greater than 1.69 mmol/L or medication use for hyperlipidemia; (4) high-density lipoprotein cholesterol (HDL-C) less than 1.03 mmol/L in men or 1.29 mmol/L in women; (5) fasting glucose equal to or greater than 5.56 mmol/L or medication use for diabetes.

### 2.3. Blood Analysis

Venous blood samples were collected after at least eight hours of fasting. Serum glucose, total cholesterol, HDL-C, low-density lipoprotein cholesterol, and triglycerides were measured by an automatic spectrophotometric assay (HITACHI 7250, Tokyo, Japan). Fasting insulin levels were assessed by a microparticle enzyme immunoassay using an AxSYM system (Abbott Laboratories, Dainabot Co., Tokyo, Japan). The homeostatic model assessment of insulin resistance (HOMA-IR) was applied to calculate the estimated degree of insulin resistance (HOMA-IR = fasting insulin × fasting plasma glucose/22.5, with glucose presented in mmol/L and insulin presented in mU/L) [18]. Serum Zn, Cu, and Fe were measured using inductively coupled plasma mass spectrometry. Serum samples were diluted 1:21 by the gravimetric method. The diluent for the sample solution and working standard was a mixture of 0.4% weight/volume tetrabutylammonium hydroxide (Sigma-Aldrich™, St. Louis, MO, USA), 0.1% weight/volume Triton X-100, and 0.1% weight/volume ethylenediaminetetraacetic acid (Sigma-Aldrich™, St. Louis, MO, USA). The Total Quant analysis method was used to establish the working range. Concentrations of Zn, Cu, and Fe in the calibration curve were 5–100 μg/L. The limit of detection for blanks and the limit of quantification for the serum samples were 0.08 µg/L and 0.78 µg/L for Cu, 0.13 µg/L and 1.65 µg/L for Zn, and 0.25 µg/L and 3.85 µg/L for Fe, respectively. Accuracy was checked against Seronorm Trace Element Human Serum (batch 704121; Nycomed AS, Oslo, Norway) as a reference material [19].

### 2.4. Statistical Analysis

Participants were divided into quartiles according to the serum concentrations of Zn, Cu, or Fe. Data are presented as the mean ± SD for continuous variables and number (percentage) for categorical variables. Multivariate logistic regression analyses were performed to estimate the odds ratio of having MetS and each metabolic factor among the quartiles of Zn, Cu, and Fe after adjustment for age, sex, current smoking, current drinking, exercise habit, body mass index (BMI), and HOMA-IR. The least square means were computed by general linear models to estimate marginal means of the serum Zn, Cu, and Fe concentrations in relation to the number of metabolic factors. Statistical analyses were performed using SPSS statistical software (V.17, SPSS, Chicago, IL, USA). A *p*-value of <0.05 was considered to be statistically significant.

## 3. Results

The basic characteristics of the participants are shown in Table 1. The average age of the participants was 65.7 ± 9.8 years in the MetS group and 66.0 ± 10.3 in the non-MetS group. The mean serum concentrations of Zn, Cu, and Fe were 941.91 ± 333.63 μg/L, 1043.45 ± 306.36 μg/L, and 1246.83 ± 538.13 μg/L, respectively.

Continuous variables are presented as the mean ± SD, and categorical variables are presented as the percentage of participants (%). *p*-values are according to the Chi-square test for categorical variables and t-test for continuous variables.

The crude odds ratios (ORs) of MetS for the highest versus the lowest quartile were 16.28 (95% CI: 10.44–25.41; *p* for trend < 0.001) for Zn, 3.50 (95% CI: 2.46–4.98; *p* for trend < 0.001) for Cu, and 6.97 (95% CI: 4.63–10.48; *p* for trend < 0.001) for Fe. After adjustment for age, sex, current smoking, current drinking, exercise habit, BMI, and HOMA-IR, the ORs of MetS for the highest versus the lowest quartile were 5.83 (95% CI: 3.35–10.12; *p* for trend < 0.001) for Zn, 2.02 (95% CI: 1. 25–3.25; *p* for trend: 0.013) for Cu and 2.11 (95% CI: 1.24–3.62; *p* for trend: 0.021) for iron, as shown in Table 2.

In Table 3, after adjustment for age, sex, current smoking, current drinking, exercise habit, BMI, and HOMA-IR, serum levels of Zn were positively associated with elevated BP (OR: 1.19, 95%, CI: 1.05–1.35, *p* = 0.006), elevated fasting glucose (OR: 2.17, 95%, CI: 1.80–2.62, *p* < 0.001), and elevated triglycerides (OR: 1.45, 95%, CI: 1.27–1.66, *p* < 0.001); serum levels of Cu were positively associated with elevated fasting glucose (OR: 1.42, 95%, CI: 1.21–1.67, *p* < 0.001) and low HDL-C (OR: 1.13, 95%, CI: 1.01–1.28, *p* < 0.041);serum levels of Fe were positively associated with elevated fasting glucose (OR: 1.72, 95%, CI: 1.43–2.07, *p* < 0.001) and elevated triglycerides (OR: 1.18, CI: 1.03–1). The LS means (±SDs) of the serum Zn, Cu, and Fe concentrations in relation to the number of metabolic factors are shown in Figure 1. The serum Zn, Cu, and Fe concentrations increased as the number of metabolic factors rose after adjusting for age, sex, current smoking, current drinking, exercise habit, BMI, and HOMA-IR (*p* for trend < 0.001)

## 4. Discussion

The study demonstrated that there were positive associations between the serum levels of Zn, Cu, and Fe and the risk of MetS. Respectively, there were 5.83-fold, 2.02-fold, and 2.11-fold risks of having MetS in the highest quartile compared with the lowest quartile of Zn, Cu, and Fe. Although adjustment for BMI, HOMA-IR, and other confounders diminished most of the magnitude of the effects, there were nonlinear increasing trends of having MetS with the escalation of Zn, Cu, and Fe levels (*p* for trend < 0.05). These findings supported that insulin resistance and obesity are major pathological components of MetS and are causes or consequences between serum gradients of Zn, Cu, and Fe and MetS. However, the persistence of direct relations between serum levels of Zn, Cu, and Fe and the risk of having MetS implied that unidentified confounding variables and unsolved pathways bypassing insulin resistance and obesity might affect this association.

Zn could have a protective role in humans by regulating inflammation, reducing oxidative stress, and being involved in lipid and glucose metabolism [20]. Moreover, Zn is known to be crucial for the synthesis, storage, and release of insulin and is related to diabetes and MetS [3]. Conversely, the hypothesized protective role of Zn has not been well demonstrated in human studies. Dietary intake [12] or supplementation [21] of Zn seemed to regress the progression of new-onset DM, whereas serum Zn had no association with DM in meta-analyses [9,10,11]. Besides, total zinc intake from both diet and supplementation failed to demonstrate the protective role toward diabetes. [10,22]. Similarly, oral Zn supplementation could not improve diabetic neuropathy, oxidative stress, or vascular function in patients with type 2 diabetes [23,24]. Additionally, studies related to Zn and MetS are scarce and inconclusive. Both the IMMIDIET study in Europe and the National Health and Nutrition Examination Survey (NHANES) in Korea failed to demonstrate an association between serum Zn levels and MetS [17,25]. Similarly, in a Chinese nested case-control study, the serum concentration of Zn was not associated with an increased risk of MetS after a three-year follow-up [16]. Only the NHANES in Korea revealed a positive association between triglycerides and serum Zn in men [17]. In contrast, our study found a strong positive association between Zn and MetS, suggesting a disturbance of zinc homeostasis in MetS patients beyond insulin resistance and obesity. Our study further clarified that Zn was positively associated with elevated fasting glucose, BP, and triglycerides after adjustment for BMI and insulin resistance. Altered Zn homeostasis has been observed in 3T3-L1 adipocytes, in which high intracellular zinc levels possess oxidative toxicity in a dose-dependent manner [26], likely through zinc transporters and metallothionein gene expression [27,28]. In our hypothesis, excessive Zn concentrations could be harmful, but the cutoff value and underlying mechanism are unclear.

Cu is another essential trace mineral that interacts with various enzymes that catalyze redox reactions [29]. Through the activity of copper/zinc superoxide dismutase, Cu facilitates the clearance of free radicals and is thought to be beneficial for chronic inflammation and is considered to be related to MetS [30]. Supported by two cross-sectional studies, dietary Cu, as a protective factor, was negatively associated with MetS [12,31]. However, cross-sectional studies in China and Korea [17,25] as well as a prospective nested cohort [16] revealed negative findings between serum Cu and MetS. Only a cross-sectional study in Lebanon showed a positive association between HDL-C and serum Cu [32]. Lacking large studies, controversial evidence suggests that Cu might act as a pro-oxidant and an antioxidant, and both excessive and deficient levels of Cu induce toxicity and cell damage [33]. In agreement with our study, an elevated serum level of Cu was associated with an increased risk of MetS and positively correlated with elevated fasting glucose and low HDL-C. Our study also showed that excessive Cu could be harmful by oxidative stress and other underlying mechanisms.

The association between Fe and DM is consistent. Elevated serum iron [34] and ferritin [9] as well as dietary Fe [16] were associated with an increased risk of DM. Additionally, excessive serum iron and ferritin were observed in the MetS group [15,35]. The pathophysiology was related to dysmetabolic iron overload characterized by altered regulation of iron transport associated with steatosis, insulin resistance, and chronic inflammation [15,36]. In line with our study, elevated serum Fe was strongly correlated with MetS. New in our study, was that we found a persistent correlation after adjusting for insulin resistance and BMI (Table 2), especially in elevated fasting glucose and triglycerides (Table 3), so we proposed that there could be another undefined pathway not mentioned before.

There are several limitations in this study. Based on the case-control design, we were not able to demonstrate the causal relationship between serum concentrations of Cu, Zn, and Fe and MetS. Although we collected and adjusted for probable confounders in this study, residual effects could still exist in constant variables, implying unmeasured and undefined factors. For example, we did not record the amount of daily micronutrient supplementation and personal dietary habits where bias independent from MetS might exist. Additionally, we did not report the duration of elevated BP, glucose intolerance, or dyslipidemia in MetS and non-MetS patients, implying potential influences of the trajectory of cardiometabolic diseases on altering serum levels of micronutrients over time. Furthermore, we used HOMA-IR to estimate the degree of insulin resistance, an indirect approach instead of accurate dynamic techniques such as a euglycemic clamp. Nonetheless, this is the first human study with a large sample size and strict study design to demonstrate a comprehensive dose-response correlation between Zn, Cu, and Fe and MetS as well as each metabolic factor. Further investigational and interventional studies are needed to elucidate the causal relationship between Zn, Cu, and Fe and MetS.

Although micronutrients evoke much concern in diabetes and MetS today, there has been inadequate evidence for their net roles in either protective or detrimental directions. Our study demonstrated that higher concentrations of Zn, Cu, and Fe have higher risks of having MetS, which has never been reported before. The dose-response relationship and probable pathways bypassing insulin resistance and obesity were also notable findings. The underlying mechanisms need further investigation.

## Figures and Tables

**Figure 1 nutrients-13-00548-f001:**
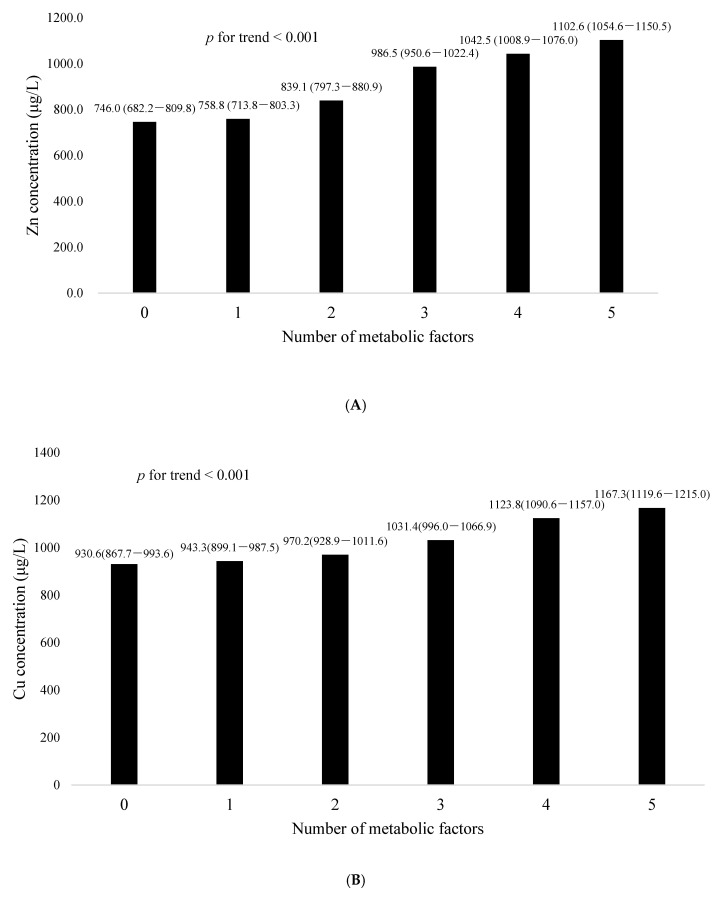
Comparison of serum concentrations of (**A**) zinc, (**B**) copper, and (**C**) iron in relation to the number of metabolic factors. The least square means of serum zinc, copper, and iron increased with the increase in the number of metabolic factors after adjusting for age, sex, current smoking, current drinking, and exercise habit. (*p* for trend < 0.001).

**Table 1 nutrients-13-00548-t001:** Characteristics of the study population in non-MetS and MetS groups.

	Non-MetS	MetS	*p*-Value
	*n* = 446	*n* = 709	
Sex			0.001
Female (%)	312 (70.0)	430 (60.6)	
Male (%)	134 (30.0)	279 (39.4)	
Age (years)	66.0 ± 10.3	65.7 ± 9.8	0.623
Weight (kg)	56.4 ± 9.1	68.7 ± 14.0	<0.001
BMI (kg/m^2^)	22.6 ± 2.7	26.8 ± 4.2	<0.001
WC (cm)	79.4 ± 8.2	91.1 ± 9.9	<0.001
Systolic BP	122.4 ± 14.8	134.4 ± 14.3	<0.001
Diastolic BP	73.8 ± 9.5	77.7 ± 10.2	<0.001
TCHO (mmol/L)	5.3 ± 0.9	4.8 ± 1.1	<0.001
Triglycerides (mmol/L)	1.1 ± 0.5	1.9 ± 1.2	<0.001
HDL-C (mmol/L)	1.5 ± 0.3	1.2 ± 0.3	<0.001
LDL-C (mmol/L)	3.1 ± 0.7	2.8 ± 0.8	<0.001
Glucose (mmol/L)	5.6 ± 1.2	7.1 ± 2.1	<0.001
Insulin (U/mL)	6.01 ± 4.20	12.69 ± 8.39	<0.001
HOMA-IR	1.56 ± 1.30	4.12 ± 3.25	<0.001
Smoking (%)	36 (8.1)	102 (14.4)	0.002
Drinking (%)	42 (9.4)	110 (15.5)	0.01
Exercise (%)	320 (71.7)	411 (58.1)	<0.001
Copper (µg/L)	949.5 ± 253.3	1101.2 ± 322.5	<0.001
Zinc (µg/L)	774.7 ± 247.4	1044.9 ± 339.3	<0.001
Iron (µg/L)	1051.6 ± 403.6	1370.0 ± 577.7	<0.001
Metabolic factors	1.25 ± 0.75	3.84 ± 0.74	<0.001

Abbreviations: BMI: body mass index; WC: waist circumference; BP: blood pressure; TCHO: total cholesterol; HDL-C: high-density lipoprotein cholesterol; LDL-C: low-density lipoprotein cholesterol; HOMA-IR: homeostasis model assessment of insulin resistance.

**Table 2 nutrients-13-00548-t002:** Odds ratios of having MetS derived from multivariate logistic regression analyses in quartiles of serum zinc, copper, and iron levels.

	Quartile of Zinc Levels				
	Q1 (*n* = 292) (≤687)	Q2 (*n* = 290) (688–871)	Q3 (*n* = 292) (872–1140)	Q4 (*n* = 291) (>1140)	*p* for Trend
MetS	98 (33.6)	159 (55.4)	197 (67.9)	255 (89.2)	
Model 1	1.00	2.46 (1.76–3.44) **	4.19 (2.97–5.93) **	16.28 (10.44–25.41) **	<0.001
Model 2	1.00	2.37 (1.69–3.34) **	4.03 (2.83–5.75) **	15.16 (9.59–23.96) **	<0.001
Model 3	1.00	1.92 (1.30–2.84) *	3.17 (2.11–4.75) **	11.02 (6.66–18.22) **	<0.001
Model 4	1.00	1.69 (1.10–2.59) *	2.06 (1.30–3.27) *	5.83 (3.35–10.12) **	<0.001
	**Quartile of Copper Levels**				
	**Q1 (*n* = 293)** **(≤821)**	**Q2 (*n* = 290)** **(822–1012)**	**Q3 (*n* = 291)** **(1013–1224)**	**Q4 (*n* = 291)** **(>1224)**	***p* for Trend**
MetS	136 (46.7)	171 (59.2)	184 (64.3)	218 (75.4)	
Model 1	1.00	1.65 (1.19–2.29) *	2.06 (1.47–2.87) **	3.50 (2.46–4.98) **	<0.001
Model 2	1.00	1.56 (1.11–2.18) *	1.95 (1.38–2.76) **	3.39 (2.35–4.87) **	<0.001
Model 3	1.00	1.53 (1.03–2.26) *	1.75 (1.17–2.62) *	2.65 (1.74–4.04) *	<0.001
Model 4	1.00	1.57 (1.01–2.44) *	1.29 (0.82–2.03)	2.02 (1.25–3.25) *	0.013
	**Quartile of Iron Levels**				
	**Q1 (*n* = 290)** **(≤900)**	**Q2 (*n* = 289)** **(901–1124)**	**Q3 (*n* = 291)** **(1125–1458)**	**Q4 (*n* = 293)** **(>1458)**	***p* for Trend**
MetS	138 (47.8)	144 (50.3)	173 (60.3)	252 (86.6)	
Model 1	1.00	1.09 (0.79–1.52)	1.64 (1.18–2.28) *	6.97 (4.63–10.48) **	<0.001
Model 2	1.00	1.06 (0.76–1.47)	1.55 (1.10–2.18) *	6.64 (4.32–10.21) **	<0.001
Model 3	1.00	0.94 (0.64–1.38)	1.31 (0.88–1.95)	5.03 (3.10–8.16) **	<0.001
Model 4	1.00	0.88 (0.57–1.36)	0.96 (0.61–1.50)	2.11 (1.24–3.62) *	0.021

Model 1: no adjustment. Model 2: adjusted for age, sex, current smoking, current drinking, and exercise habit. Model 3: adjusted for variables in Model 2, plus BMI as a confounding factor. Model 4: adjusted for variables in Model 3, plus HOMA-IR as a confounding factor. Abbreviation: MetS, metabolic syndrome; HOMA-IR, homeostasis model assessment of insulin resistance. * For *p* < 0.05. ** For *p* < 0.001.

**Table 3 nutrients-13-00548-t003:** Odds ratios of having individual metabolic factors derived from multivariate logistic regression analyses in quartiles of serum zinc, copper, and iron levels.

	Elevated WC		Elevated BP		Elevated Glucose		Elevated		Low HDL–C	
							Triglycerides			
	OR (95% CI)	*p* for Trend	OR (95% CI)	*p* for Trend	OR (95% CI)	*p* for Trend	OR (95% CI)	*p* for Trend	OR (95% CI)	*p* for Trend
Zinc										
Model 1	1.54 (1.38–1.72)	<0.001	1.35 (1.21–1.50)	<0.001	3.05 (2.64–3.53)	<0.001	1.81 (1.62–2.03)	<0.001	1.16 (1.04–1.29)	0.006
Model 2	1.63 (1.45–1.84)	<0.001	1.34 (1.20–1.50)	<0.001	2.87 (2.46–3.34)	<0.001	1.78 (1.58–2.01)	<0.001	1.24 (1.11–1.39)	<0.001
Model 3	1.32 (1.10–1.58)	0.002	1.24 (1.11–1.40)	<0.001	2.61 (2.23–3.06)	<0.001	1.65 (1.45–1.86)	<0.001	1.12 (0.99–1.26)	0.066
Model 4	1.15 (0.95–1.40)	0.163	1.19 (1.05–1.35)	0.006	2.17 (1.80–2.62)	<0.001	1.45 (1.27–1.66)	<0.001	1.04(0.91–1.18)	0.59
Copper										
Model 1	1.39 (1.25–1.55)	<0.001	1.12 (1.01–1.24)	0.032	1.58 (1.41–1.78)	<0.001	1.24 (1.11–1.38)	<0.001	1.29 (1.16–1.43)	<0.001
Model 2	1.36 (1.22–1.52)	<0.001	1.14 (1.02–1.27)	0.017	1.63 (1.44–1.85)	<0.001	1.25 (1.12–1.39)	<0.001	1.26 (1.13–1.41)	<0.001
Model 3	1.19 (1.00–1.42)	0.047	1.08 (0.97–1.20)	0.172	1.55 (1.35–1.76)	<0.001	1.18 (1.05–1.32)	0.004	1.19 (1.07–1.34)	0.002
Model 4	1.09 (0.90–1.31)	0.376	1.04 (0.93–1.17)	0.501	1.42 (1.21–1.67)	<0.001	1.10 (0.98–1.25)	0.121	1.13 (1.01–1.28)	0.041
Iron										
Model 1	1.38 (1.24–1.54)	<0.001	1.19 (1.07–1.32)	0.001	2.31 (2.03–2.64)	<0.001	1.49 (1.34–1.67)	<0.001	1.08 (0.97–1.20)	0.155
Model 2	1.47 (1.31–1.66)	<0.001	1.18 (1.06–1.32)	0.004	2.19 (1.91–2.52)	<0.001	1.46 (1.29–1.64)	<0.001	1.15 (1.02–1.28)	0.02
Model 3	1.31 (1.09–1.57)	0.004	1.11 (0.99–1.24)	0.087	2.08 (1.79–2.41)	<0.001	1.36 (1.21–1.54)	<0.001	1.05 (0.93–1.19)	0.402
Model 4	1.13 (0.93–1.38)	0.207	1.07 (0.95–1.22)	0.263	1.72 (1.43–2.07)	<0.001	1.18 (1.03–1.35)	0.016	0.94 (0.83–1.07)	0.362

Model 1: no adjustment. Model 2: adjusted for age, sex, current smoking, current drinking, and exercise habit. Model 3: adjusted for variables in Model 2, plus BMI as a confounding factor. Model 4: adjusted for variables in Model 3, plus HOMA-IR as a confounding factor. Abbreviation: MetS, metabolic syndrome; HOMA-IR, homeostasis model assessment of insulin resistance.

## Data Availability

The rights to the data belong to our research group. Selected variables without personal identification codes could be provided after application.

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
