# Peer review of "Association of Serum Levels of Zinc, Copper, and Iron with Risk of Metabolic Syndrome"

_nutrients, 2021, doi:10.3390/nu13020548_

Round 1
Reviewer 1 Report
In my opinion, the paper is interesting, but the graphic form is not the best. In addition, the article is very valuable and can be printed.
Author Response
1. In my opinion, the paper is interesting, but the graphic form is not the best. In addition, the article is very valuable and can be printed
Response: Thanks for your kind commendation. To make the figure clearer, we have added micronutrient units at the vertical axis and add number of metabolic factors in the horizontal axis of Figure 1 on Page 6-7.

Reviewer 2 Report
Authors presented a case-study entitled "Association of Serum Levels of Zinc, Copper and Iron with Risk of Metabolic Syndrome" that overall satisfied the reviewer expectations. The manuscript was well presented, with a very good english level and easy to follow. the reviewer detected 2 minor error:
abstract: in the sentence "Therefore, we conduct a case control study to explored the relationship between serum levels of zinc..." should be explore instead of explored
section 2.1:the sentence "Personal habit of smoking and drinking were defined as binary categories Current smokers were defined as those smoking for more than 6 months prior to participating in this study and labeled as ”1”." it lacks a dot before current
results discussion of table 3: in the sentence "...BMI and HOMA-IR (P for trend<0.0" the rest of the numbers from the p value are missing
the reviewer also acknowledge the methodology used as well as the discussion provided. the reviewer only could suggest to include in the discussion a little bit more literatura regarding the use of some of these elements in the treatment of some metabolic diseases, for example, the use of zinc supplementation in type 2 diabetes patients in order to give even more soundness and importance to the study presented in this manuscript.
Author Response
Authors presented a case-study entitled "Association of Serum Levels of Zinc, Copper and Iron with Risk of Metabolic Syndrome" that overall satisfied the reviewer expectations. The manuscript was well presented, with a very good English level and easy to follow. The reviewer detected 2 minor error:
1. Abstract: in the sentence "Therefore, we conduct a case control study to explored the relationship between serum levels of zinc..." should be explore instead of explored. section 2.1:the sentence "Personal habit of smoking and drinking were defined as binary categories Current smokers were defined as those smoking for more than 6 months prior to participating in this study and labeled as ”1”." it lacks a dot before current. Results discussion of table 3: in the sentence "...BMI and HOMA-IR (P for trend<0.0" the rest of the numbers from the p value are missing
Response: We have to apologize for the typos and missing words. We have corrected “explored” to explore, added a dot before current and put on the P for trend<0.001(page 5, line 3).
2. The reviewer also acknowledge the methodology used as well as the discussion provided. The reviewer only could suggest to include in the discussion a little bit more literature regarding the use of some of these elements in the treatment of some metabolic diseases, for example, the use of zinc supplementation in type 2 diabetes patients in order to give even more soundness and importance to the study presented in this manuscript.
Response: Thanks for your comment. We have amended discussion section as follows:” Besides, total zinc intake from both diet and supplementation failed to demonstrate the protective role toward diabetes. [10, 22]. Similarly, oral Zn supplementation could not improve diabetic neuropathy, oxidative stress or vascular function in patients with type 2 diabetes [23, 24]….” in the 2nd paragraph on Page 8.

Reviewer 3 Report
The authors should describe in more detail the mode of selection of the subjects in the sample.
They should also raise their hypothesis to better explain their findings in the discussion.
Author Response
1. The authors should describe in more detail the mode of selection of the subjects in the sample.
Response: Thanks for your comment. We have added the mode of selection in Method section as follows “…They were invited for the health and nutrition survey in the outpatient clinic if they could comply with our study protocol…” in the first paragraph on Page 2.
2. They should also raise their hypothesis to better explain their findings in the discussion.
Response: In discussion section, we have already mentioned in the original manuscript and highlighted as follows:“In our hypothesis, excessive Zn concentrations could be harmful, but the cutoff value and underlying mechanism are unclear.” in the 2nd paragraph on Page 8. Furthermore, we have added “Our study also showed that excessive Cu could be harmful by oxidative stress and other underlying mechanisms.” in the 3rd paragraph on Page 8. Finally, “ What else and new in our study was that we found a persistent correlation after adjusting for insulin resistance and BMI, especially in elevated fasting glucose and triglycerides, proposed another undefined pathway not mentioned before.” has been stated in the first paragraph on Page 9 of the original manuscript.
